# Spatiotemporal Transformer Neural Network for Time-Series Forecasting

**DOI:** 10.3390/e24111651

**Published:** 2022-11-14

**Authors:** Yujie You, Le Zhang, Peng Tao, Suran Liu, Luonan Chen

**Affiliations:** 1College of Computer Science, Sichuan University, Chengdu 610065, China; 2Key Laboratory of Systems Biology, Hangzhou Institute for Advanced Study, University of Chinese Academy of Sciences, Hangzhou 310024, China; 3Key Laboratory of Systems Health Science of Zhejiang Province, Hangzhou Institute for Advanced Study, University of Chinese Academy of Sciences, Hangzhou 310024, China; 4State Key Laboratory of Cell Biology, Institute of Biochemistry and Cell Biology, Center for Excellence in Molecular Cell Science, Chinese Academy of Sciences, Shanghai 200031, China; 5Guangdong Institute of Intelligence Science and Technology, Hengqin, Zhuhai 519031, China; 6West China Biomedical Big Data Center, Med-X Center for Informatics, West China Hospital, Sichuan University, Chengdu 610041, China

**Keywords:** time-series, spatiotemporal information transformation, attention mechanism, transformer network

## Abstract

Predicting high-dimensional short-term time-series is a difficult task due to the lack of sufficient information and the curse of dimensionality. To overcome these problems, this study proposes a novel spatiotemporal transformer neural network (STNN) for efficient prediction of short-term time-series with three major features. Firstly, the STNN can accurately and robustly predict a high-dimensional short-term time-series in a multi-step-ahead manner by exploiting high-dimensional/spatial information based on the spatiotemporal information (STI) transformation equation. Secondly, the continuous attention mechanism makes the prediction results more accurate than those of previous studies. Thirdly, we developed continuous spatial self-attention, temporal self-attention, and transformation attention mechanisms to create a bridge between effective spatial information and future temporal evolution information. Fourthly, we show that the STNN model can reconstruct the phase space of the dynamical system, which is explored in the time-series prediction. The experimental results demonstrate that the STNN significantly outperforms the existing methods on various benchmarks and real-world systems in the multi-step-ahead prediction of a short-term time-series.

## 1. Introduction

Time-series forecasting is a critical ingredient in many fields, such as computational biology [1,2], finance [3], traffic flow [4], and geoscience [5]. However, due to the limited measurement conditions, we usually can only obtain short-term time-series samples [6]. On one hand, since a short-term dataset has no sufficient information, it becomes a challenging task to carry out accurate multi-step-ahead prediction using a short-term time-series. On the other hand, we can measure high-dimensional data in many real-world systems, which include rich information of the dynamics on the target variable and thus can be exploited to compensate the insufficiency of the short-term data. However, there is the curse of dimensionality in effectively analyzing and predicting high-dimensional time-series [7]. As an empirical example, Figure 1 shows the prediction results on the 64-dimentional pendulum datasets from fewer observed time-series steps (50 steps, Figure 1a) to enough observed time-series steps (100 steps, Figure 1b). Figure 1c shows the forecasting metric variation with observed time-series steps. When there are fewer observed data, the NRMSE shows unsatisfactory performance, and the prediction model fails.

With decades of development, generally, there are two major types of methods for time-series forecasting. One type is model-based methods, which consist of autoregression (AR) [8], autoregressive integrated moving average (ARIMA) [9], and support vector regression (SVR) [10,11,12]. AR and ARIMA are mostly used in univariate regression analysis, because the vector AR model used for multivariate prediction requires a large number of parameters, resulting in low prediction accuracy with a small training dataset. However, SVR also requires a large training dataset for the time-series prediction, so it is hard to accurately estimate the parameters of the model-based methods with short-term time-series. The other type is neural networks based on deep learning methods [13,14,15], such as recurrent neural networks (RNNs) [16], long short-term memory (LSTM) networks [17], and reservoir computing [18]. Because they usually require a large training dataset to learn the nonlinear characteristics of the dynamical system to infer the temporal evolution of variables, it is often necessary to introduce dimension reduction or additional a priori knowledge to reconstruct their dynamic or statistical patterns.

To explore high-dimensional information [19], the spatiotemporal information (STI) transformation equation [7] has recently been developed based on the delay embedding theorem [20]. As a set of nonlinear equations, the STI equation transforms the spatial information of high-dimensional variables into the future temporal information of any target variable, thus equivalently expanding the sample size and alleviating the short-term data problem [19]. Based on the STI equation, previous studies employed randomly distributed embedding (RDE) [7] and an anticipated learning machine (ALM) [19] to fit the STI equation. However, the robustness and accuracy of the prediction are not satisfactory due to the difficulty in solving the nonlinear STI equation with high dimensions and multiple parameters.

Recently, the transformer neural network [21] has been developed as an extension of neural networks based on autoencoder frameworks [22,23], and it is suitable for sequential information processing. Unlike sequence-aligned models [24], the transformer processes an entire sequence of data and leverages self-attention mechanisms to learn information in the sequence, which allows us to model the relationship of variables without considering their distance in the input sequences. In particular, since the attention mechanism can not only fully capture the global information but also focus on the important content [21], it can alleviate the curse of dimensionality with great potentiality [19].

To overcome the problems in time-series prediction, we propose a spatiotemporal transformer neural network (STNN) for efficient multi-step-ahead prediction of high-dimensional short-term time-series by taking the advantages of both the STI equation and the transformer structure. Here, we summarize our contributions as follows:An STNN is developed to adopt the STI equation, which transforms the spatial information of high-dimensional variables into the temporal evolution information of one target variable, thus equivalently expanding the sample size and alleviating the short-term data problem.A continuous attention mechanism is developed to improve the numerical prediction accuracy of the STNN.A continuous spatial self-attention structure in the STNN is developed to capture the effective spatial information of high-dimensional variables, with the temporal self-attention structure used to capture the temporal evolution information of the target variable, and the transformation attention structure used to combine spatial information and future temporal information.We show that the STNN model can reconstruct the phase space of the dynamical system, which is explored in the time-series prediction.

The rest of this study is organized as follows. Section 2 mainly describes the relevant works on the spatiotemporal transformation equation and transformer neural network for time-series prediction. Section 3 presents the overall STNN architecture and describes the relevant theory and procedures. Section 4 shows the computational experiments on various benchmarks and real-world systems. Finally, we present our conclusion and discuss directions of future study.

## 2. Related Works

### 2.1. Delay Embedding for Spatiotemporal Transformation Equation

For a general discrete time dynamical system [25], Equation (1) defines the dynamical evolution of its state.
(1)$$X^{t+1}=\phi \left(X^{t}\right)$$

Xt=(x1t,x2t,…,xDt)′ are defined in a D-dimensional space at time step t, where the symbol ′ means the transpose of a vector. The map ϕ: ℝD→ℝD is a nonlinear function, which pushes states from time t to time t+1. 

To bridge the spatial information and the temporal evolution information, we let $Y^{t}=\left(y^{t},y^{t+1},\ldots ,{\;y}^{t+L-1}\right)\prime =\left(x_{\mathrm{target}}^{t},x_{\mathrm{target}}^{t+1},\ldots ,x_{\mathrm{target}}^{t+L-1}\right)\prime$, which are the values of one target variable selected from X for (L-1)-step-ahead prediction with L > 1. Note that $X^{t}$ is spatial/high-dimensional information due to the multiple (D) variables at one time point t, while $Y^{t}$ is temporal information due to the single variable at multiple (L) time points. When the system of Equation (1) is in a steady state or in a manifold V with dimension d, based on Takens’ embedding theorem [20,26], we can construct the following spatiotemporal information (STI) transformation equation, which maps the D-dimensional data $X^{t}$ to L-dimensional data $Y^{t}$.
(2)$$\Phi \left(X^{t}\right)=Y^{t}=(y^{t},y^{t+1},\ldots ,{\;y}^{t+L-1}{)}^{\prime }$$
where, generally, D >> L and L > 2d. Clearly, the spatiotemporal information (STI) transformation equation transforms the available/previous spatial information $X^{t}$ of multiple variables to the future temporal information $Y^{t}$ of one target variable at each time point t [7]. For the prediction, the studies of [7,19] indicated that there are L sub-predictors acting on each dimension. If the measured time-series has M time steps, we can rewrite Equation (2) in a matrix form, as shown in Equation (3).



(3)
$$\left[\begin{array}{cccc} \Phi_{1}(X^{1}) & \Phi_{1}(X^{2}) & \cdots & \Phi_{1}(X^{M})\\ \Phi_{2}\left(X^{1}\right) & \Phi_{2}\left(X^{2}\right) & \cdots & \Phi_{2}\left(X^{M}\right)\\ \vdots & \vdots & \ddots & \vdots \\ \Phi_{L}(X^{1}) & \Phi_{L}(X^{2}) & \cdots & \Phi_{L}(X^{M}) \end{array}\right]=\left[\begin{array}{cccc} y^{1} & y^{2} & \cdots & y^{M}\\ y^{2} & y^{3} & \cdots & {\;\hat{y}}^{M+1}\\ \vdots & \vdots & \ddots & \vdots \\ y^{L} & y^{L+1} & \cdots & {\;\hat{y}}^{M+L-1} \end{array}\right]$$



Since the observation variables are up to time step M, the ^ indicates that the values of target variable y from time steps M+1 to M+L−1 need to be predicted in addition to the maps $\Phi_{i}\;\mathrm{for}\;i=1,\ldots ,L$, given $X^{t}$ for t = 1, …, M. Thus, we can have (L-1)-step-ahead prediction of a target variable y by solving $\Phi_{i}$ and $Y^{t}$ of Equation (3), provided that $X^{t}$ for t = 1, …, M are available. Generally, even if the dimension D of the original system is very high, the dimension d of its steady state or manifold is very low for most real-world systems, i.e., D >> d. Thus, we generally choose a small d by letting L = 2d+1 in the computation of Equation (3).

Several works have tried to predict high-dimensional short-term time-series with the STI equation. For example, Ma et al. [7] first constructed the STI transformation equation with a computational framework, named randomly distributed embedding (RDE), for one-step-ahead prediction of short-term time-series. The novelty of this RDE framework is rooted in exploiting the information embedded in many low-dimensional non-delay attractors as well as in the appropriate use of the distribution of the target variable for prediction. Chen et al. [27] developed an auto-reservoir computing framework, named the auto-reservoir neural network (ARNN), to approximate the nonlinear STI equation to a linear-like form, which can efficiently carry out multi-step-ahead prediction based on a short-term high-dimensional time-series. Such ARNN transformation equivalently expands the sample size, but its linear-like approximation sacrifices the accuracy to some extent, although it has potential in practical applications of artificial intelligence.

### 2.2. Transformer Neural Network for Time-Series Prediction

The transformer has been widely used in the field of natural language processing, which is described in detail by Vaswani et al. [21]. Unlike sequence-aligned models [24], the transformer processes an entire sequence of data and leverages the classical self-attention mechanism to capture global dependencies of the sequence X, as shown in Equation (4): (4)$$\mathrm{Attention}\left(Q,K,V\right)=\mathrm{Softmax}(\frac{{\mathrm{QK}}^{T}}{\sqrt{d_{k}}}\cdot \mathrm{Mask})V$$
where the query matrix ${Q=\mathrm{XW}}_{Q}$, key matrix ${K=\mathrm{XW}}_{K}$, and value matrix ${V=\mathrm{XW}}_{V}$ are transformed by X; $W_{Q}$, $W_{K}$, and $W_{V}$ are learnable parameter matrices; and $d_{k}$ means the dimension of matrix K. Note that a mask matrix is applied to filter out rightward attention to avoid future information leakage by setting all upper triangular elements in ($\frac{{\mathrm{QK}}^{T}}{\sqrt{d_{k}}}$) to −∞.

However, at present, the transformer structure has not been well studied for processing high-dimensional short-term time-series data. Moreover, only a few studies consider the effective modeling of time-series from the perspective of the attention mechanism. For example, Shih et al. [28] proposed an attention mechanism to extract temporal patterns, and it successfully captures the temporal information of time-series. Moreover, the attention mechanism will select the variables that are helpful for forecasting. Therefore, the vector of the result finally obtained through the attention is a weighted sum containing the information across multiple time steps, and it has potential for time-series prediction by reducing the unrelated variables.

## 3. Problem Setup and Methodology

### 3.1. Problem Definition

Given a set of observed high-dimensional short-term time-series data X=(X1,X2,….,Xt,…,XM)∈ℝD×M, M represents the observed time-series steps and D represents the variable dimension. We define the state at any time step t as Xt=(x1t,x2t,…,xDt)′, t = 1,2,...,M. We aim to have (L-1)-step-ahead prediction of a target variable (y = x_target_) based on the time-series **X**, i.e., to predict (yM+1,yM+2,…, yM+L-1)=(xtargetM+1,xtargetM+2,…,xtargetM+L-1), where xtargett is the target variable which is any one among D variables of Xt. 

This study’s aim is to construct a neural network model for the prediction, the input of which is the observed D-dimensional variables $X^{t}$ and the L-dimensional target variables ${\bar{Y}}^{t}=(0,y^{t},y^{t+1},...,y^{t+L-2}{)}^{\prime }$ at any time step t, and the output of which is the one-step-ahead prediction ($y^{t+L-1}$). We show how to construct such a model in Section 3.2 in detail. Therefore, under a rolling forecast with a fixed window size L, our goal is to implement (L-1)-step-ahead prediction and eventually output the final (L-1)-step-ahead prediction result of the target variable $\left({\hat{y}}^{M+1},...,{\hat{y}}^{M+L-1}\right)$.

### 3.2. STNN Model

This study proposes a model named STNN to realize the spatiotemporal information transformation. The STNN aims to efficiently solve the nonlinear STI transformation equation, Equation (5), by exploring the transformer, i.e., construct $\Phi =\left[\Phi_{1}{,\Phi }_{2}{,\ldots ,\Phi }_{L}\right]’$, which is a smooth diffeomorphism mapping [26].
(5)$$\left[\Phi \left(\begin{array}{c} \begin{array}{c} x_{1}^{1}\\ x_{2}^{1} \end{array}\\ \begin{array}{c} \vdots \\ x_{D}^{1} \end{array} \end{array}\right)\Phi \left(\begin{array}{c} \begin{array}{c} x_{1}^{2}\\ x_{2}^{2} \end{array}\\ \begin{array}{c} \vdots \\ x_{D}^{2} \end{array} \end{array}\right)\ldots \Phi \left(\begin{array}{c} \begin{array}{c} x_{1}^{M}\\ x_{2}^{M} \end{array}\\ \begin{array}{c} \vdots \\ x_{D}^{M} \end{array} \end{array}\right)\right]=\left[\begin{array}{cccc} y^{1} & y^{2} & \cdots & y^{M}\\ y^{2} & y^{3} & \cdots & y^{M+1}\\ \vdots & \vdots & \ddots & \vdots \\ y^{L} & y^{L+1} & \cdots & y^{M+L-1} \end{array}\right]$$

D represents the variable dimension, L represents the embedded dimension, and M represents the observed time-series steps.
(6)$$\Phi \left(X^{t},{\bar{Y}}^{t}\right)=\mathrm{Decoder}\left(\mathrm{Encoder}\left(X^{t}\right),{\bar{Y}}^{t}\right)={\hat{Y}}^{t}$$

The STNN model (Figure 2) employs the STI transformation equation (Equation (5)) with two specific transformer modules to carry out multi-step-ahead prediction. As the description of Equation (6), one of the modules is the encoder, which takes D-dimensional variables at the same time t ($X^{t}$) as inputs. Then, the encoder extracts effective spatial information from the input variables. After that, the effective spatial information is transferred to the decoder. The other is the decoder, which inputs an L-1-length time-series from the target variable Y (${\bar{Y}}^{t}$). Then, the decoder extracts the temporal evolution information of the target variable. After that, the decoder predicts the future values of the target variable (${\hat{Y}}^{t}$) by combining the spatial information of the input variables ($X^{t}$) and the temporal information of the target variable (${\bar{Y}}^{t}$).

Note that y in Y is also one variable among the measured variables X. Φ in Equation (6) is not exactly the same as that in Equation (5) due to ${\bar{Y}}^{t}$, but Φ can be expressed in a similar form using an appropriate mathematical implementation. Clearly, the nonlinear STI transformation Φ is solved by the encoder–decoder pair. Similar to the classical seq2seq framework [29], ${\bar{Y}}^{t}=(0,y^{t},y^{t+1},...,y^{t+L-2}{)}^{\prime }$ is an L-dimensional time-series, which is formed by replacing the first dimension of $Y^{t-1}{=(y}^{t-1}{,y}^{t}{,\ldots ,\;y}^{t+L-2}{)}^{\prime }$ with zero, thus keeping the causality of the prediction. Next, we detail the encoder and decoder modules.

#### 3.2.1. Encoder

The encoder is composed of two layers. One is a fully connected layer, and the other is a continuous spatial self-attention layer. We employ the continuous spatial self-attention layer to extract the effective spatial information from the high-dimensional input variables $X^{t}$. 

The fully connected layer is used to obtain the effective expression by smoothing the input high-dimensional variables Xt and filtering the noise, which is a forward propagation network composed of a layer of neurons described by Equation (7).
(7)XFFNt=ELU(WFFNXt+bFFN)
where FFN stands for feedforward neural network, WFFN∈ℝD×D is the coefficient matrix, bFFN∈ℝD is the bias, and ELU is the activation function.

The continuous spatial self-attention layer takes $X_{\mathrm{FFN}}^{t}$ as an input. Since the self-attention layer takes high-dimensional variables at the same time as inputs, the encoder can extract the spatial information from the input variables. In order to obtain the effective spatial information (${\overset{´}{\mathrm{SSA}}}^{t}$), we propose a continuous attention mechanism for the spatial self-attention layer instead of the classical discrete probability-based attention mechanism [21]. The left of Figure 2 shows our continuous attention mechanism, whose procedure can be described as follows.

Firstly, we generate three training weight matrices, $W_{E}^{Q}$, $W_{E}^{K}$, and $W_{E}^{V}$, for the continuous spatial self-attention layer. 

Secondly, Equation (8) computes the query matrix ($Q_{E}^{t}$), key matrix ($K_{E}^{t}$), and value matrix ($V_{E}^{t}$) for the continuous spatial self-attention layer by multiplying the output $X_{\mathrm{FFN}}^{t}$ of the fully connected layer by the above three weight matrices for time step (t).
(8)$$\{\begin{array}{c} Q_{E\;}^{t}{=X}_{\mathrm{FFN}}^{t}W_{E}^{Q}\\ K_{E\;}^{t}{=X}_{\mathrm{FFN}}^{t}W_{E}^{K}\\ \begin{array}{c} V_{E}^{t}{=X}_{\mathrm{FFN}}^{t}W_{E}^{V} \end{array} \end{array}$$

Thirdly, Equation (9) executes the matrix dot product to obtain the expression of key spatial information (${\overset{´}{\mathrm{SSA}}}^{t}$) for the input variables $X^{t}$.
(9)$${\overset{´}{\mathrm{SSA}}}^{t}=\exp(\frac{1}{\sqrt{d_{E}}}{\cdot Q}_{E}^{t}{\cdot K}_{E}^{t}{}^{\prime }{)\cdot V}_{E}^{t}$$
where $d_{E}$ is the dimension of the query matrix ($Q_{E}^{t}$), key matrix ($K_{E}^{t}$), and value matrix ($V_{E}^{t}$). Different from the classical discrete probability-based attention mechanism [21], the continuous attention mechanism (Equation (9)) can guarantee a smooth data transmission for the encoder. 

Fourthly, we compute the normalized expression of effective spatial information (${\mathrm{SSA}}^{t}$) using residual join and the layer normalization operation [21] (Equation (10)), which can prevent the gradient from quickly disappearing and accelerate the model convergence speed.
(10)$${\mathrm{SSA}}^{t}=\mathrm{Norm}\left(X_{\mathrm{FFN}}^{t}+{\overset{´}{\mathrm{SSA}}}^{t}\right)$$

#### 3.2.2. Decoder

The decoder combines effective spatial and temporal evolution information, and it consists of two fully connected layers, i.e., one continuous temporal self-attention layer and one transformation attention layer. 

As shown in Figure 2, we obtain the effective expression (${\bar{Y}}_{\mathrm{FFN}}^{t}$) after filtering the noise of the input data (${\bar{Y}}^{t}$) using a fully connected layer. Next, we send the output (${\bar{Y}}_{\mathrm{FFN}}^{t}$) into the continuous temporal self-attention layer. The continuous temporal attention layer focuses on the historical temporal evolution information among different time steps of the target variable (${\bar{Y}}^{t}$). Because the impact on time is irreversible, we determine the current state of the time-series using historical information but not future information. Therefore, the continuous temporal attention layer uses a masked attention mechanism [21] to screen out future information. The detailed procedure is as follows. 

Firstly, we generate three training weight matrices, $W_{D}^{Q}$, $W_{D}^{K}$, and $W_{D}^{V}$, for the temporal spatial self-attention layer. 

Secondly, Equation (11) computes the query matrix ($Q_{D}^{t}$), key matrix ($K_{D}^{t}$), and value matrix ($V_{D}^{t}$) for the temporal spatial self-attention layer.



(11)
$$\{\begin{array}{c} Q_{D\;}^{t}{=\bar{Y}}_{\mathrm{FFN}}^{t}W_{D}^{Q}\\ K_{D\;}^{t}{=\bar{Y}}_{\mathrm{FFN}}^{t}W_{D}^{K}\\ \begin{array}{c} V_{D}^{t}{=\bar{Y}}_{\mathrm{FFN}}^{t}W_{D}^{V} \end{array} \end{array}$$



Thirdly, Equation (12) executes the matrix dot product to obtain the expression of the temporal evolution information (${\overset{´}{\mathrm{TSA}}}^{t}$) for the input variable (${\bar{Y}}^{t}$).
(12)$${\overset{´}{\mathrm{TSA}}}^{t\;}=\exp(\frac{1}{\sqrt{d_{D}}}\cdot Q_{D}^{t}\cdot K_{D}^{t}{}^{\prime }{\cdot \mathrm{Mask})\cdot V}_{D}^{t}$$
where $d_{D}$ is the dimension of the query matrix ($Q_{D}^{t}$), key matrix ($K_{D}^{t}$), and value matrix ($V_{D}^{t}$) for the temporal spatial self-attention layer. Additionally, we employ Equation (13) to describe the mask matrix with $d_{M}$ dimension.
(13)$$\mathrm{Mask}={\left[\begin{array}{ccccc} 1 & 0 & 0 & \cdots & 0\\ 1 & 1 & 0 & \cdots & 0\\ 1 & 1 & 1 & \cdots & 0\\ \vdots & \vdots & \vdots & \ddots & 0\\ 1 & 1 & \cdots & 1 & 1 \end{array}\right]}_{d_{M}{\times d}_{M}}$$

By setting zero in the mask matrix (Equation (13)), we prevent each position from attending to the coming positions to capture the historical temporal evolution information of the target variable. 

Fourthly, we compute the normalized expression of the temporal evolution information (${\mathrm{TSA}}^{t}$) using residual join and the layer normalization operation [21].
(14)$${\mathrm{TSA}}^{t}=\mathrm{Norm}\left(Y_{\mathrm{FFN}}^{t}+{\overset{´}{\mathrm{TSA}}}^{t}\right)$$

Fifthly, the continuous transformation attention layer combines the effective spatial information (${\mathrm{SSA}}^{t}$) and the temporal evolution information (${\mathrm{TSA}}^{t}$) to predict the future values of the target variable (${\overset{´}{\mathrm{TA}}}^{t}$) using Equation (15). Here, $d_{{\mathrm{SSA}}^{t}}$ is the dimension of ${\mathrm{SSA}}^{t}$.
(15)$${\overset{´}{\mathrm{TA}}}^{t}=\frac{1}{\sqrt{d_{{\mathrm{SSA}}^{t}}}}{\mathrm{TSA}}^{t}{\cdot \mathrm{SSA}}^{t}{}^{\prime }\cdot {\mathrm{SSA}}^{t}$$

Lastly, we put ${\mathrm{TA}}^{t}$ into residual join, the layer normalization operation, and a fully connected layer in a proper order to compute the L-dimensional prediction result ${\hat{Y}}^{t}$.

(16)$$\{\begin{array}{c} {\mathrm{TA}}^{t}=\mathrm{Norm}\left({\mathrm{TSA}}^{t}+{\overset{´}{\mathrm{TA}}}^{t}\right)\\ {\hat{Y}}^{t}=\mathrm{ELU}(W\cdot {\mathrm{TA}}^{t}+b) \end{array}$$ where W is the coefficient matrix, b is the bias, and ELU is the activation function. 

#### 3.2.3. Objective Function for STNN Model

The STNN framework defines the objective function (Equation (17)) to minimize the loss ε.

(17)$$\min\;\varepsilon =\overset{M-L+1}{\underset{t=1}{\sum }}{\left\|{\hat{Y}}^{t}-Y^{t}\right\|}_{2}^{2}+\lambda {\left\|W\right\|}_{2}^{2}$$ where M represents the observed time-series steps, L represents the length of the fixed window size, and ${\hat{Y}}^{t}$ and $Y^{t}$ are the predicted and true values of a target variable, respectively. ${\left\|\cdot \right\|}_{2}$ is the Frobenius norm, λ controls the importance of the penalty, and W is the parameter space of the STNN.

## 4. Experiments

This section evaluates the performance of the STNN framework on several high-dimensional short-term time-series datasets.

### 4.1. Datasets

We empirically performed multi-step-ahead prediction using a short-term high-dimensional time-series on six datasets, including two benchmarks and four public datasets from real-world systems.

#### 4.1.1. Benchmarks

**Pendulum:** The nonlinear pendulum [30] is a classic textbook example of dynamical systems, which is used for benchmarking models [31,32]. We generated a nonlinear pendulum dataset with 80 observed time-series steps (M = 80), and we mapped the series {x^t^} to a high-dimensional space via a random orthogonal transformation to obtain the 64-dimensional training snapshots (D = 64). The training dataset is composed of the first 63 steps, and the remaining 17 steps are for the testing dataset. 

**Lorenz:** The Lorenz system [33] is a meteorological dynamic system for studying essential dynamical characteristics of nonlinear systems, which is used in chaotic time-series prediction [34]. This study generated a 90-dimensional coupled Lorenz dataset (D = 90) with 80 observed time-series steps (M = 80). The training dataset is composed of the first 61 steps, and the remaining 19 steps are for the testing dataset. Short-term prediction on the Lorentz system is helpful to verify the prediction performance of the model on the chaotic system.

#### 4.1.2. Public Datasets

**Traffic Speed (TS):** The traffic speed (mile/h) dataset was collected from 207 loop detectors (D =207) on Highway 134 of Los Angeles County [35]. We employed the STNN to predict the traffic flow with 80 observed time-series steps (M = 90). The training dataset is composed of the first 71 steps, and the remaining 19 steps are for the testing dataset. Short-term prediction on traffic speed datasets is helpful to detect the running speed of vehicles and reduce the occurrence of traffic accidents.

**Gene:** The gene expression data [36] were obtained from rats, and some important genes are related to the circadian rhythm, which is a fundamentally important physiological process regarded as the “central clock” of mammals. Here, we used the data measured by an Affymetrix microarray on a laboratory rat with 84 genes (D = 84) and 22 observed time-series steps (M = 22) by creating a record every 2 h. The training dataset is composed of the first 16 steps, and the remaining 6 steps are for the testing dataset. Short-term prediction on the circadian rhythm gene datasets is helpful to understand whether the physiological rhythm in the organism is disordered in advance and ensure life and health.

**Solar:** The data were originally collected from Wakkanai, Japan [37]. Here, we used solar irradiance datasets based on 450 observed time-series steps (M = 450) from 51 sampling sites (D = 51). Since 2011, the 51 sampling sites have formed a system to reflect the changes in solar irradiance by creating a record every 10 min. The training dataset is composed of the first 301 steps of solar irradiance, and the remaining 149 steps are for the testing dataset. Short-term prediction on the solar irradiance datasets is essential to minimize energy costs and provide a high power quality [38].

**Traffic Flow (TF):** The data were originally collected from the California Department of Transportation and describe the road occupancy rate of the Los Angeles County highway network [39]. Here, we used a subset of the dataset, which contains 228 sensors (D = 228) with 40 observed time-series steps (M = 40) for each sensor. The training dataset is composed of the first 33 steps, and the remaining 7 steps are for the testing dataset. Short-term prediction on the traffic flow datasets is helpful to understand the traffic jam and relieve the traffic pressure during peak traffic hours.

### 4.2. Experimental Details

Here, we briefly summarize the basics; more details on the network components and setups are provided in the Appendix A. 

**Platform:** All the classical methods used in this study and our STNN framework are based on the deep learning framework PyTorch. The experimental hardware environment was configured with an Intel(R) Core (TM) i7-4710HQ CPU @ 2.50GHz, with 8.0 GB of memory. 

**Baselines:** We selected six classical time-series forecasting methods and the STNN* to compare the performance with our STNN method. It should be noted that we incorporated the canonical attention mechanism in the STNN, which is named STNN*. 

The six classical time-series forecasting methods consist of autoregressive integrated moving average (ARIMA) [9], support vector regression with linear kernel (SVR) [10], support vector regression with radial basis function (RBF) [11], a recurrent neural network (RNN) [16], and the Koopman autoencoder (KAE) [22]. 

**Metrics:** We used the Pearson correlation coefficient (PCC) [40,41,42,43] and normalized root mean square error (NRMSE) [42,44,45,46] to measure the performance of each algorithm.



(18)
$$\mathrm{PCC}=\frac{\sum_{i=m}^{m+L-1}({\hat{y}}^{i}{-\hat{\mu })(y}^{i}-\mu )}{\sigma_{p}\sigma }$$


(19)
$$\mathrm{NRMSE}=\frac{\sqrt[{2}]{\frac{1}{L}\sum_{i=m}^{m+L-1}{\left\|{\hat{y}}^{i}-y^{i}\right\|}^{2}}}{\sigma }$$



The PCC and NRMSE are computed based on the last column ${\hat{y}}^{M+1},...,{\hat{y}}^{M+L-1}$ of the Y matrix in Equation (5), where ${\hat{y}}^{i}$ is the predicted value at the time step i; $\hat{\mu }$ and μ are the mean values of the prediction and true data, respectively; and $\sigma_{p}$ and σ are the standard deviation of the predicted data and true data, respectively.

### 4.3. Results and Analysis

#### 4.3.1. Time-Series Forecasting

Table 1 summarizes the evaluation results (PCC and NRMSE) for all the methods on the six datasets. We randomly selected four target variables (such as targe∈[1,2,3,4] in Equation (6)) to be predicted from each dataset, and each method recorded the average and variance of the predictions. The best average results are highlighted in boldface. The last row of Table 1 records the number of times each method obtained the best metric. 

From Table 1, we can observe the following. (1) The proposed STNN model improves the inference performance (winning counts in the last row) across all datasets. This proves that STNN has better performance than existing methods in alleviating short-term data problems. (2) The prediction variance of STNN is kept at a small level on all datasets, which indicates that STNN has a relatively stable prediction ability compared with existing methods. (3) The STNN beats its canonical degradation STNN* (the STNN model with the canonical attention mechanism in the transformer) mostly in winning counts, i.e., 9 > 2, which supports the fact that the continuous attention mechanism can efficiently improve the numerical prediction accuracy of the STNN. (4) The STNN model returned significantly better results than ARIMA and SVR. This reveals that the STNN can effectively transform the spatial information of high-dimensional variables into the future temporal information to acquire a better prediction capacity than the classical time-series algorithms.

#### 4.3.2. Characteristic Experiment

**Robustness:** To test the robustness of the STNN model, we increased the noise strength (σ) in the pendulum data and explored the change in prediction accuracy (detailed in Appendix A). Figure 3 shows the change in the PCC and NRMSE with the noise strength (σ) from 0 to 0.5 in the pendulum data for five different methods. In Figure 3a, compared with the other four methods, the STNN not only has a maximum PCC value with the increase in noise strength, but also maintains the PCC value at a high level. In Figure 3b, compared with the other four methods, the STNN always has the lowest NRMSE value with the increase in noise strength. This demonstrates that the STNN has the strongest anti-noise ability, and its prediction has the greatest accuracy under strong noise.

**Embedded dimension:** Generally, Takens’ embedding theorem [26] demonstrates that time delay embedding is topologically equivalent to the unknown phase space of dynamical systems, when the embedded dimension L > 2d+1. Therefore, we constructed different time delay embedding STI functions with the embedded dimension L from 2 to $L_{\max}$ (Figure 4a) to investigate the optimum range of the embedded dimension by performing experiments on the pendulum dataset.

Figure 4b summarizes the forecasting metrics of four randomly selected target predicted variables (such as targe∈[1,2,3,4] in Equation (6)) and the mean metrics of the four target predicted variables. According to the frequency variation in the mean metrics ($\frac{\Delta \mathrm{NRMSE}}{\Delta L}$), we empirically divided the embedded dimension into three areas (I, II, and III). In area I, the NRMSE value decreases rapidly with the increase in the embedded dimension. In area II, the NRMSE value reaches its minimum and remains at a low level with the increase in the embedded dimension. In area III, the NRMSE value increases with the increase in the embedded dimension.

To explore why the NRMSE decreases first and then increases with the increase in the embedded dimension, we repeated the above experiments on the pendulum dataset with data points [50,60,...,100] using the STNN.

Figure 5a intuitively presents the two embedded dimension points. With the increase in the observed time-series steps, the coordinates of the three embedded points form lines 1 and 2, which divide the embedded dimension into three areas (I, II, and III).

Figure 5b and Table 2 intuitively present the mean and variance of the NRMES corresponding to three areas. Through analysis of variance (nonparametric Kruskal–Wallis test) [47] (Table 2), the *p*-value (1.89 × 10^−3^) shows statistically significant differences in the NRMSE errors among the three areas. This proves that the prediction accuracy of the STNN will be greatly improved with the increase in the embedded dimension in area I. Then, the best prediction effect will be obtained when the embedded dimension reaches area II. The prediction accuracy of the STNN framework will decrease when we increase the embedded dimension in area III.

Figure 6 details line 1 and line 2 to show why the NRMSE decreases first and then increases with the embedded dimension.

(1) As the observed time-series steps increase, Figure 6a shows that the division points of area I and area II remain approximately horizontal. Therefore, we calculated the mean of the division points as the interception of line 1, which equals 5.67. Here, d = 2 is the manifold dimension of pendulum. Since L = 5.67 is close to 2d + 1 = 5, this proves that the STNN cannot make an accurate prediction when the embedded dimension L is close to 2d + 1 (in area I).

(2) As the observed time-series steps increase, Figure 6b shows an obvious positive correlation with the observed time-series steps, which implies that when the embedded dimension increases, the STNN prediction accuracy is negatively related to the observed time-series steps. Therefore, this proves that the prediction accuracy is decreased due to the lower amount of training data and longer prediction length with the increase in the embedded dimension (in area III).

#### 4.3.3. The Performance of STNN

In order to explore how STNN’s performance will depend on M (observed time-series steps), D (dimension of time-series), and L (embedded dimension), we conducted experiments on two variables from the randomly selected variables in Section 4.3.1 with pendulum data, solar data, and TS data. The details of the experiment are recorded in the Appendix A.

Figure 7 details the forecasting metric (NRMSE) variation with the observed time-series steps (M). As the observed time-series steps increase, Figure 7 shows that the NRMSE is decreasing on pendulum data, solar data, and TS data. Therefore, we can conclude that more observed data can provide more temporal information for STNN prediction, thus improving the prediction accuracy of STNN.

Figure 8 details the forecasting metric (NRMSE) variation with the dimension of time-series (D). As the dimension of time-series increases, Figure 8 shows that the NRMSE remains almost unchanged on pendulum data, solar data, and TS data. Therefore, we can conclude that STNN can achieve high-performance prediction results with less spatial information by mining the correlation between high-dimensional system variables.

Figure 9 details the forecasting metric (NRMSE) variation with the embedded dimension (L). As the embedded dimension increases, Figure 9 shows that the NRMSE decreases first and then increases with the increase in the embedded dimension on pendulum data, solar data, and TS data, which is consistent with the previous discussion in Section 4.3.2.

#### 4.3.4. Ablation Experiment

We also conducted additional ablation experiments.

The performance of continuous spatial self-attention and temporal self-attention: To verify that the continuous spatial attention mechanism and continuous temporal attention mechanism are indispensable in the STNN, we removed the two components while keeping other settings unchanged. In Table 2, STNN^#^ uses a fully connected layer instead of the continuous spatial self-attention mechanism, and STNN^##^ uses a fully connected layer instead of the continuous temporal self-attention mechanism. In Table 3, the predicted results show that the STNN has better performance than the counterparts (STNN^#^ and STNN^##^), which demonstrates that the continuous spatial self-attention and continuous temporal self-attention can improve the prediction accuracy of time-series. Therefore, we consider that continuous spatial self-attention and continuous temporal self-attention are critical for the spatiotemporal transformation of STI.

## 5. Conclusions

To solve two problems in predicting high-dimensional short-term time-series—the lack of sufficient information and the curse of dimensionality for high-dimensional short-term time-series prediction—this study proposed the STNN framework by taking the advantages of both the STI transformation equation and the transformer neural network framework to accurately predict future values of a short-term time-series in a multi-step-ahead manner. 

By comparing the STNN prediction with the existing methods on various benchmarks and real-world systems, we conclude that the STNN not only can significantly improve the accuracy and robustness of the prediction, but it also has strong generalization ability. Additionally, as the dimension of time-series increases, the NRMSE remains almost unchanged, which shows that we may improve the operation efficiency of STNN through the dimension reduction method and maintain high prediction performance.

## Figures and Tables

**Figure 1 entropy-24-01651-f001:**
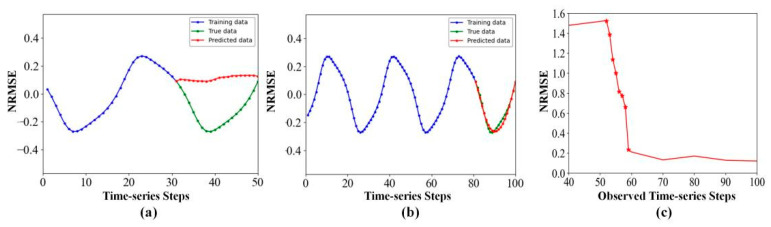
(**a**) Short-term dataset without sufficient information. (**b**) Long-term dataset with sufficient information. (**c**) The prediction ability of existing models fails in the case of fewer observed time-series steps.

**Figure 2 entropy-24-01651-f002:**
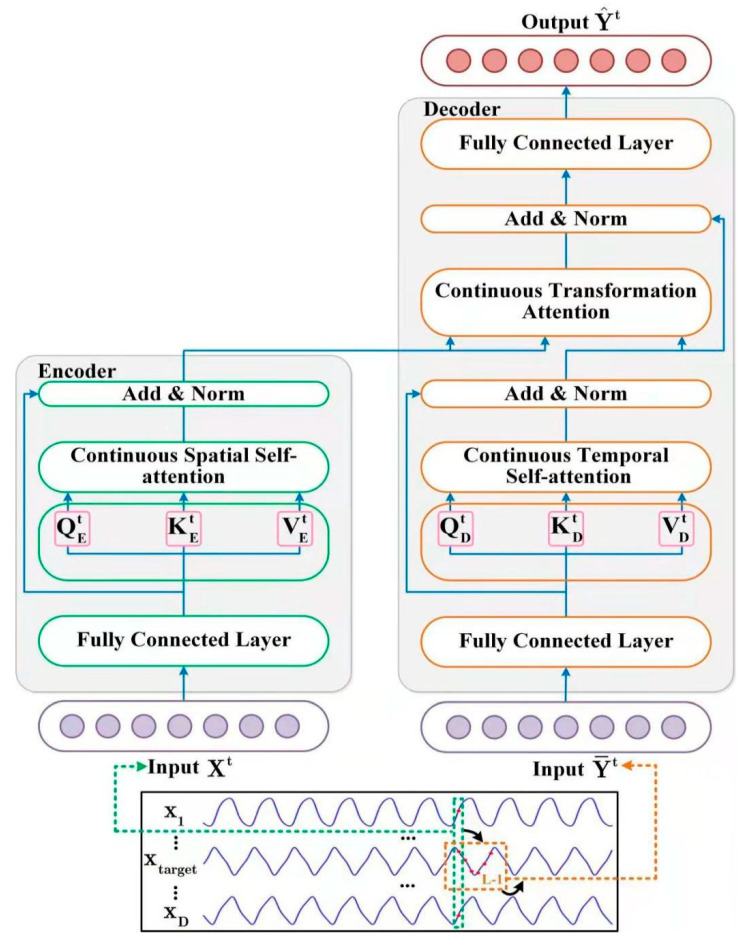
Overview of the proposed STNN framework. In the left model, the encoder receives D-dimensional series inputs $X^{t}$ and outputs the spatial information feature to the transformation attention layer of the decoder. In the right model, the decoder receives L-dimensional series inputs ${\bar{Y}}^{t}$ with the spatial information feature from the encoder and outputs the L-dimensional prediction result ${\hat{Y}}^{t}$, where ${\bar{Y}}^{t}=(0,y^{t},y^{t+1},...,y^{t+L-2}{)}^{\prime }$ is an L-dimensional series, which is formed by an L-1-length series ${{(y}^{t}{,y}^{t+1}{,\ldots ,\;y}^{t+L-2})}^{\prime }$ from the observed times series with the first dimension filling out zero.

**Figure 3 entropy-24-01651-f003:**
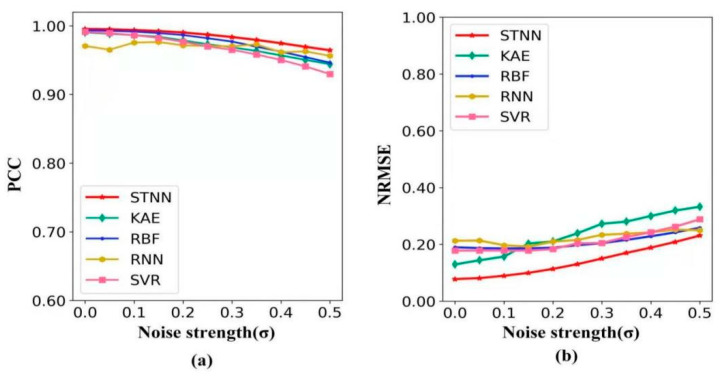
The robustness of five methods (STNN, KAE, RBF, RNN and SVR). (**a**) The forecasting metric PCC variation with noise strength. (**b**) The forecasting metric NRMSE variation with noise strength.

**Figure 4 entropy-24-01651-f004:**
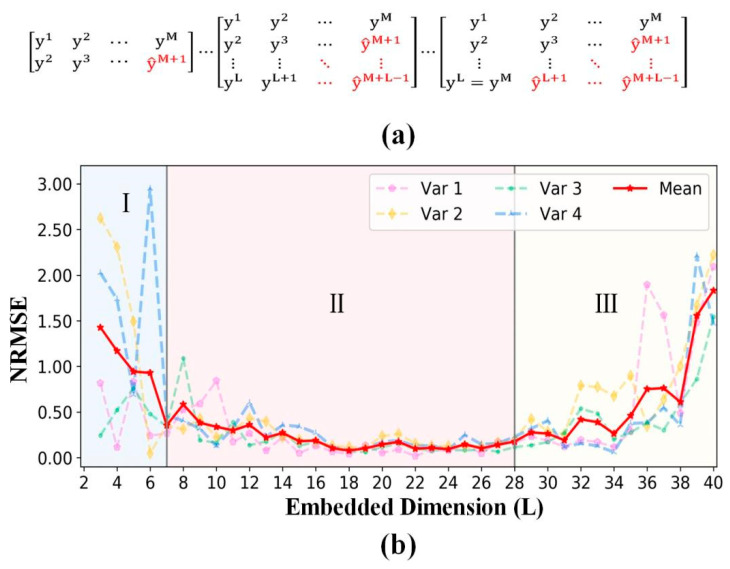
(**a**) The STI function varying with the embedded dimension from 2 to $L_{\max}$. (**b**) Forecasting metric variation with the embedded dimension. Var 1, 2, 3, and 4 are four randomly selected target predicted variables from Equation (6). The red line presents the mean metrics of the four target predicted variables.

**Figure 5 entropy-24-01651-f005:**
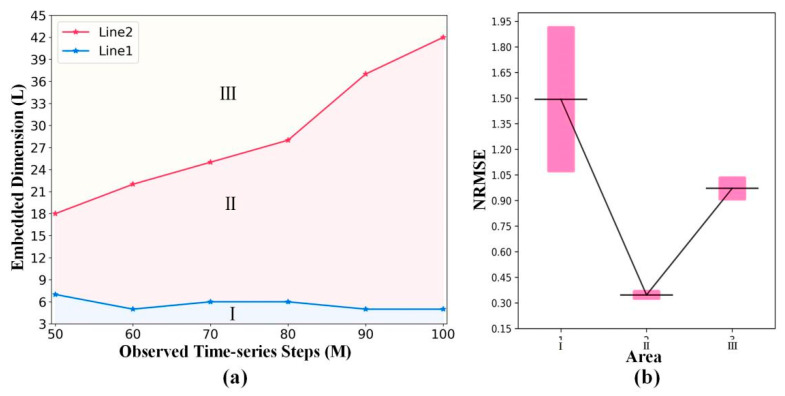
(**a**) Connection between the embedded dimension and observed time-series steps. For each observed time-series step M, the embedded dimension of the points on line 1 corresponds to the location on the x-axis of the first vertical line in Figure 4a, which divides the embedded dimension into area I and area II. Similarly, the embedded dimension of the points on line 2 corresponds to the location on the x-axis of the second vertical line in Figure 4a, which divides the embedded dimension into area II and area III. (**b**) The mean and variance of the NRMES corresponding to three areas.

**Figure 6 entropy-24-01651-f006:**
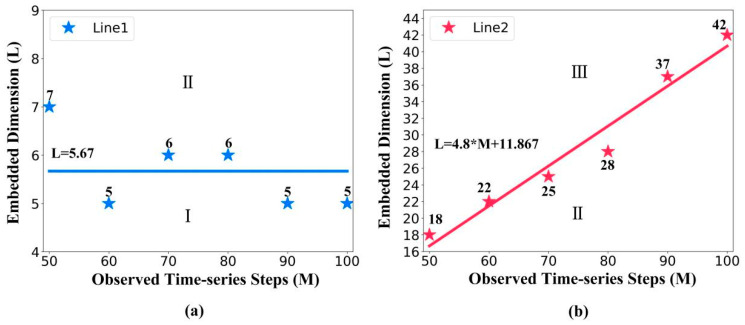
(**a**) The basic information of line 1. The division points between area I and area II generated an approximately horizontal line, line 1. The interception of line 1 equals 5.67. (**b**) The basic information of line 2. The division points between area II and area III generated line 2 that shows an obvious positive correlation with the observed time-series steps.

**Figure 7 entropy-24-01651-f007:**
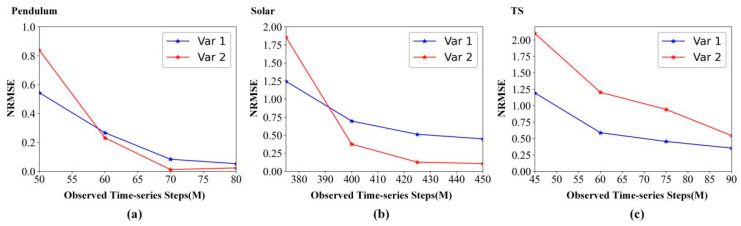
Forecasting metric variation with the observed time-series steps (M). (**a**) Pendulum data. (**b**) Solar data. (**c**) TS data.

**Figure 8 entropy-24-01651-f008:**
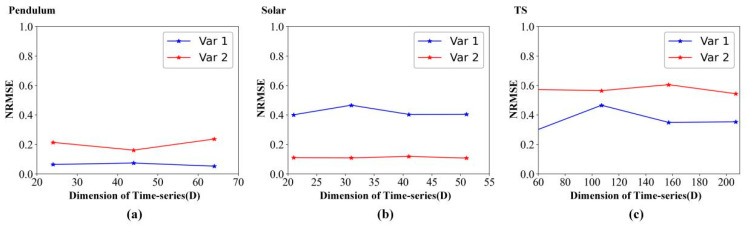
Forecasting metric variation with the dimension of time-series (D). (**a**) Pendulum data. (**b**) Solar data. (**c**) TS data.

**Figure 9 entropy-24-01651-f009:**
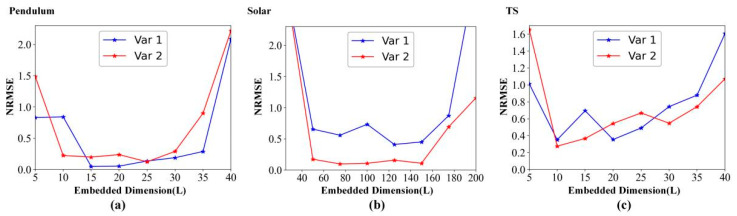
Forecasting metric variation with the embedded dimension (L). (**a**) Pendulum data. (**b**) Solar data. (**c**) TS data.

**Table 1 entropy-24-01651-t001:** Time-series forecasting results on six datasets by seven methods.

Dataset	Metric		STNN	STNN*	ARIMA	SVR	RBF	RNN	KAE
Pendulum	PCC	Mean	**0.994**	0.884	0.371	0.991	0.993	0.947	0.990
		Var	1.419 × 10^−5^	0.018	0.248	3.250 × 10^−5^	1.250 × 10^−6^	8.065 × 10^−4^	8.475 × 10^−5^
	NRMSE	Mean	**0.146**	0.590	0.679	0.178	0.190	0.258	0.129
		Var	0.005	0.028	0.051	0.010	0.014	3.482 × 10^−4^	1.725 × 10^−5^
Lorenz	PCC	Mean	**0.995**	−0.554	0.906	−0.306	−0.446	0.308	−0.525
		Var	3.569 × 10^−5^	0.194	0.013	0.640	0.601	0.254	0.245
	NRMSE	Mean	**0.097**	2.451	0.620	1.580	1.600	1.816	2.629
		Var	0.002	0.781	0.833	0.294	0.133	0.184	0.786
Gene	PCC	Mean	0.395	0.381	0.243	0.404	**0.446**	0.171	−0.065
		Var	0.007	0.115	0.160	0.087	0.014	0.383	0.162
	NRMSE	Mean	**0.658**	1.058	0.948	0.762	1.017	1.110	1.948
		Var	0.005	0.125	0.0416	0.037	0.038	0.213	0.270
TS	PCC	Mean	**0.866**	0.668	0.258	0.514	0.545	0.198	−0.223
		Var	0.005	0.102	0.149	0.022	0.009	0.089	0.164
	NRMSE	Mean	**0.504**	0.755	1.082	1.226	1.303	1.232	1.275
		Var	0.011	0.090	0.074	0.022	0.049	0.108	0.151
Solar	PCC	Mean	0.948	**0.951**	0.188	0.643	0.831	0.155	0.010
		Var	0.001	0.001	0.112	0.065	3.747 × 10^−4^	0.005	0.046
	NRMSE	Mean	0.372	**0.345**	1.129	0.809	0.934	1.580	1.602
		Var	0.024	0.019	0.005	0.058	0.007	0.091	0.096
TF	PCC	Mean	**0.989**	0.846	0.821	0.987	0.990	0.507	0.658
		Var	2.497 × 10^−4^	0.003	0.092	4.262 × 10^−4^	3.168	0.712	0.313
	NRMSE	Mean	**0.121**	0.787	0.334	0.380	1.362	0.802	6.793
		Var	0.002	0.058	0.100	0.025	0.185	0.136	1.825
Winning counts			**9**	2	0	0	1	0	0

**Table 2 entropy-24-01651-t002:** The NRMSE of three areas.

Observed Time-Series Steps (M)	Area I	Area II	Area III
50	1.5367	0.5644	0.9169
60	1.3138	0.3799	0.9085
70	2.8385	0.5307	1.3988
80	1.1171	0.2452	0.6116
90	1.4030	0.2434	0.7835
100	0.7451	0.1138	1.2030
Mean	1.4924	0.3462	0.9704
Variance	0.4255	0.0263	0.0680
Variance analysis	*p*-value = 1.89 × 10^−3^		

**Table 3 entropy-24-01651-t003:** Ablation of continuous spatial self-attention and temporal self-attention.

Model	Metric	Pendulum	Lorenz
STNN	PCC	0.9983	0.9979
	NRMSE	0.0778	0.0967
STNN^#^	PCC	0.9955	0.7362
	NRMSE	0.0780	0.7118
STNN^##^	PCC	0.9944	0.5703
	NRMSE	0.0802	0.9747

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
