# Peer review of "Spatiotemporal Transformer Neural Network for Time-Series Forecasting"

_entropy, 2022, doi:10.3390/e24111651_

Round 1
Reviewer 1 Report
The authors describe STNN, a method to forecast the time-series of a single target variable Y(t=M...M+L-1) L-1 steps into the future, based on high-dimensional observations \vec{X}(t=1...M) by using a transformer neural network to learn the mapping between the observations and target variable. The authors then demonstrate this method on several synthetic and real-world datasets, comparing to other forecasting approaches. The performance of STNN as a function of L and M are explored for the nonlinear pendulum.
While the idea for the method is reasonable in principle and the results appear promising, the manuscript suffers from a lack of clarity that hampers my enthusiasm. My critiques are as follows:
* The authors state that they are exploiting Taken's well-known time-delay embedding theorem, which demonstrates that a time-delay embedding of a single variable will reconstruct, up to diffeomorphism, the phase space of a d-dimensional dynamical system. As near as I can tell, they are treating their target Y(t=M...M+L-1) in the time-delay embedded space, with L embedding dimensions. It is not clear to me, then, what role Takens' theorem is playing here -- that is, whether one conceives of Y as L observations in a time-series (typical forecasting) or Y as a point in L-dimensional space (time-delay embedded with a time delay set to 1), the problem would be equivalent. That is, eq 3.3 (and onward) is simply a matrix, and it seems irrelevant whether one thinks of this as a sliding window or an embedding. I would appreciate some clarification on this matter, as I feel I must be missing something about the relevance of Takens' theorem to this paper.
* L is variously referred to as "steps ahead", "embedded dimension", and "data length" (eg in Table 2). "Steps ahead" seems clear and corresponds to the equations. "Embedded dimension" is also true if my understanding in point 1 is correct, although less interpretable than "steps ahead". "Data length" is badly confusing -- I would have expected this to be the historical (observed) data length M. It is made worse by M also being referred to as "Data length" elsewhere. Some clarification is needed.
* The authors apply STNN to multiple datasets, with X dimensionality ranging from D=1 (the pendulum) to D=228 (traffic flow sensors) and various timeseries lengths. The authors then compare the prediction performance in terms of both the correlation (PCC) with and the NRMSE from the true timeseries, for four randomly selected target variables. Averages of the PCC and NRMSE for the four targets are reported. However, several things are unclear (see following points).
* Presumably, the PCC and NRMSE are only computed on "true forecasts" (ie, the last column of the Y matrix in eq 3.3) by holding out the trailing end of the timeseries as a test set. Confirmation of this important point would be valuable.
* It is not clear how much of the data was used to fit/train the STNN models; what is M? (I believe this is given in the Supplement, but it needs to be in the main paper, and made very clear.) More critically, what is the relationship between M, D (the dimensionality of the observations), and L on the prediction performance? What happens when M & D are downsampled?
* The PCC and NRMSE are reported as averages across the predictions for four randomly selected targets. It would be useful to know how much variance there is in these metrics, however. That is -- what is the worst-case performance of STNN, and is it better/worse than the worst-case performance of the other forecasting methods? To this end, predictions for more than four targets would be much better; indeed, there is no reason not to exhaustively try to predict all the possible targets in the data.
* The datasets are not very well described / sources for the data are not given. EG, the reference for the circadian gene expression data [36] does not pertain to circadian dynamics at all, and the GEO accession number (the standard identifier for gene expression datasets) is not given. Likewise, the Lorenz system as described in the supplement mentions the addition of Gaussian noise, but not the variance of said noise. This makes it challenging for readers to reproduce the results or benchmark other methods relative to STNN.
* For the Lorentz system, are the parameters chosen such that it is in the chaotic regime? How does performance of the prediction vary as a function of the regime that the system is in?
* In the comparisons, by far the most attention is given to the pendulum, which has dimension D=2, ie position & momentum. The authors claim to predict four targets, however. Once again, I am confused -- how can four targets be predicted when only two are present? Some clarification would be helpful.
* Fig 2 -- I am not sure I understand how the noise strength was varied; as I read it, the pendulum system described in the supplement is simply a harmonic oscillator, without noise?
* Figures 3-6 should be combined, and a detailed explanatory caption given. EG, as presented it is not clear that the legend "Line2/Line1" in Fig 5 pertains to the location on the x-axis of the two vertical lines in Fig 4.
* More critically regard the above, how was the placement of those two lines determined? All that is said is "we empirically divide the embedded dimension into three areas" but the criteria are not given.
* The authors provide a "variance analysis" p-value, but do not describe the test. If this was standard parametric ANOVA, were the assumptions (normality, equality of variance) met? Or was a nonparametric (Kruskal-Wallis) test used? Perhaps more generally, it is not entirely clear what is learned by this test. The NRMSEs differ significantly across the three areas, but the areas were selected based on the NRMSE flattening out. Presumably, then, the null hypothesis (that the NRMSE variance is not associated with the area) is false and the p-value will be small by construction. I recommend just dropping this altogether.
* Figs 7/8 should also be combined, and need much more detail in their captions to explain what is shown.
* In general, I found the extensive attention to a very simple system (the pendulum) a bit unsatisfying. In general, forecasting the behavior of a deterministic, nonchaotic pendulum is not of interest; rather, the challenge we face in forecasting arises in complex (possibly chaotic) systems. For STNN to be of interest, then, the performance as a function of M, D, and L should be explored in the real-world datasets, as well as the Lorentz system.
* I found the whole manuscript a bit challenging to follow; it should be revised for clarity/English usage.
Reviewer 2 Report
Please see attached.

Round 2
Reviewer 1 Report
The new manuscript is considerably improved in readability and several of my comments have been addressed.
However, the main critique that I had -- that the authors focus on predicting the behavior of a simple harmonic oscillator but do not explore performance adequately in the higher dimensional & more unpredictable systems for which STNN is intended -- remains unaddressed.
In general, my impression is that this was a fast revision with little attention paid to the more substantial critiques. The specific points that I remain concerned about follow:
* I had written: "The PCC and NRMSE are reported as averages across the predictions for four randomly selected targets. It would be useful to know how much variance there is in these metrics, however. That is -- what is the worst-case performance of STNN, and is it better/worse than the worst-case performance of the other forecasting methods? To this end, predictions for more than four targets would be much better; indeed, there is no reason not to exhaustively try to predict all the possible targets in the data." The authors reply: "As described in the Related works L122 that our work is inspired by the following reference paper[1]. Therefore, consistent with this reference paper, we randomly selected four dimensions of high-dimensional time series for experiments."
I find this unsatisfying as I still can't tell how representative the PCC and NRMSE results are. Better statistics are crucial for readers to evaluate the method. As it stands, it is unclear whether the four targets were "lucky".
* I had written: "The datasets are not very well described / sources for the data are not given. EG, the reference for the circadian gene expression data does not pertain to circadian dynamics at all, and the GEO accession number (the standard identifier for gene expression datasets) is not given." -- this remains the case. With the current citation is impossible for the reader to find the data to reproduce the results. A reference to the data itself is needed.
* I had written: "In the comparisons, by far the most attention is given to the pendulum, which has dimension D=2, ie position & momentum. The authors claim to predict four targets, however." The authors explain in reply "The position of generated pendulum data has dimension D=2, and then we can use a random orthogonal transformation to obtain the 60-dimensional training snapshots (D=60)." [NB: in the paper it says 64.]
This raises more questions than it answers. I assume that when the paper states "we mapped the series {xt} to a high-dimensional space via a random orthogonal transformation to obtain the 64-dimensional training snapshots (D=64)" it means that the 2(position,momentum) by 80(timepoints) by data from the oscillator was multiplied by a random orthogonal matrix of size 64x2 to obtain a 64x80 matrix of "training snapshots"; or perhaps a 1(position) by 80(timepoints) data was multiplied by a random matrix of size 64x1? In either case, it is not clear what was the purpose of transforming a simple harmonic oscillator it into a 64-D timeseries. The underlying dimensionality is STILL 2-D -- ie, the rank of the 64-D transformed matrix will still be 2 (or 1, if only position was used) -- so how is it meaningful to predict four targets? They cannot possibly be independent, so how does one interpret the result? Additionally, how many periods of the oscillation do the 80 timepoints cover? It is a very different thing to predict the behavior after >1 period has been observed, when Takens' embedding will describe the full cycle, than if those 80 observations happen to cover 1/2 a cycle.
Given the amount of attention given to this system in the paper, all of this needs to be very clear.
* I had written: "In general, I found the extensive attention to a very simple system (the pendulum) a bit unsatisfying. In general, forecasting the behavior of a deterministic, nonchaotic pendulum is not of interest; rather, the challenge we face in forecasting arises in complex (possibly chaotic) systems. For STNN to be of interest, then, the performance as a function of M, D, and L should be explored in the real-world datasets, as well as the Lorentz system." In reply, the authors stand by their decision: "Since our main purpose is to prove whether the STNN model can reconstruct the phase space of the dynamic system using Takens' embedding theorem, we explore the prediction performance of the STNN model by pendulum system experiments with key parameters, M(observed time-series steps), D(dimension of time-series), and L(embedded dimension). Also, modeling chaotic system is by STNN should be an interesting study, but we have to do it in the distant future."
While it is certainly the author's prerogative to decide what falls within the scope of their research, I must say that I do not find the ability to predict the outcome of simple harmonic motion to be very compelling. In particular, the authors state that STNN is a solution to forecasting *high-dimensional* short-term time-series in a multi-step-ahead manner. This is a very important problem in many fields, and a solution to it would be highly significant! They propose that the STNN method can use information contained in the high-dimensional, but short, timeseries data to forecast any target variable. If it works, that is a very exciting result. However, I do not believe that it has been adequately demonstrated. Focusing on a deterministic system where the underlying dimensionality is at most two (and where it is unclear how short the timeseries actually is with respect to the system's natural frequency) is not a convincing demonstration that STNN performs well for the stated task, and gives the reader very little indication of how STNN's performance will depend on M(observed time-series steps), D(dimension of time-series), and L(embedded dimension) in a non-trivial use case.
It thus remains my opinion that this is promising, but much more work is needed.
Reviewer 2 Report
The authors have submitted a revised manuscript -- the English is much better, and I feel the presentation is overall a step in the right direction. This is short of a complete re-write I was imagining and the manuscript is still lacking in many respects, mostly related to math notation.
Below I (again) just give a few examples, each point on its own is trivial but taken altogether I consider this a problem. Ultimately up to the editor what level of rigour is acceptable.
- L48: "the vector AR model used for multivariate prediction requires a large number of parameters, resulting in low prediction accuracy." This is an unsubstantiated claim. Neural networks have tons of parameters, does this always result in low prediction accuracy?
- Eq (2.1): X^t should be bold. Generally there is no consistency with bold and not bold in the manuscript.
- Eq (2.2): Transpose operator missing. Sometimes ^T is also used for transpose.
- L157: "by setting all upper triangular elements to −∞," - of which matrices? All matrices? Or just those relating to Ybar?
- L170: Unless 2 < t < M, it shouldn't be in the definition of X. Similarly in L172, t=1,2,...,M,.... but earlier it was stated that M is the number of observed steps so no dots needed after M.
- L197: Has Yhat^t been defined?
- Comment on Figure 1 and general exposition: Both Ybar^t and X^t are passing through the same architecture, it is strange that the X^t part is treated as a decoder but that of Ybar^t is not.
- Generally when reading the manuscript it comes across as though you tried this architecture out and "struck gold". Statements like (again, this is just an example) "self-attention enables the transformer to focus on different aspects of temporal patterns, and thus it has potential for time-series prediction by reducing the unrelated variables" are vague and unclear. E.g., What are the mechanics behind self-attention that enable this (in this example?)
